# Significance of MnO_2_ Type and Solution Parameters in Manganese Removal from Water Solution

**DOI:** 10.3390/ijms24054448

**Published:** 2023-02-23

**Authors:** Magdalena M. Michel, Mostafa Azizi, Dorota Mirosław-Świątek, Lidia Reczek, Bogumił Cieniek, Eleonora Sočo

**Affiliations:** 1Institute of Environmental Engineering, Warsaw University of Life Science, 02-787 Warsaw, Poland; 2Institute of Materials Engineering, University of Rzeszow, 35-310 Rzeszow, Poland; 3Department of Inorganic and Analytical Chemistry, Faculty of Chemistry, Rzeszow University of Technology, 35-959 Rzeszow, Poland

**Keywords:** environmental engineering, groundwater treatment, adsorption, MnO_2_ polymorphs, akhtenskite, birnessite, cryptomelane, pyrolusite

## Abstract

A very low concentration of manganese (Mn) in water is a critical issue for municipal and industrial water supply systems. Mn removal technology is based on the use of manganese oxides (MnO_x_), especially manganese dioxide (MnO_2_) polymorphs, under different conditions of pH and ionic strength (water salinity). The statistical significance of the impact of polymorph type (akhtenskite ε-MnO_2_, birnessite δ-MnO_2_, cryptomelane α-MnO_2_ and pyrolusite β-MnO_2_), pH (2–9) and ionic strength (1–50 mmol/L) of solution on the adsorption level of Mn was investigated. The analysis of variance and the non-parametric Kruskal–Wallis H test were applied. Before and after Mn adsorption, the tested polymorphs were characterized using X-ray diffraction, scanning electron microscope techniques and gas porosimetry analysis. Here we demonstrated the significant differences in adsorption level between MnO_2_ polymorphs’ type and pH; however, the statistical analysis proves that the type of MnO_2_ has a four times stronger influence. There was no statistical significance for the ionic strength parameter. We showed that the high adsorption of Mn on the poorly crystalline polymorphs leads to the blockage of micropores in akhtenskite and, contrary, causes the development of the surface structure of birnessite. At the same time, no changes in the surfaces of cryptomelane and pyrolusite, the highly crystalline polymorphs, were found due to the very small loading by the adsorbate.

## 1. Introduction

The importance of groundwater is evidenced by the fact that 65% of water supplied for domestic uses in the EU is abstracted exactly from this source [1]. Manganese (Mn) is an unwanted element commonly occurring in groundwater, and its high-effective removal has significant meaning when water supplies municipal or industrial distribution networks. The health threshold is higher (<0.4 mg/L) than that resulting from operational issues (0.1 mg/L) [2]. Unacceptable taste and appearance of water, clogging pipes and increasing hydraulic resistance; black deposits on plumbing fitting; growth of bacteria leading to odor and biofilm; and laundry staining are the most important problems. The permissible level of Mn in drinking water varies; in 103 countries, it ranges from 0.05 to 0.5 mg/L [3]. However, it is pointed out that even an Mn concentration over 0.02 mg/L can cause these problems, and the discretionary threshold is suggested at a level of 0.01 mg/L [4].

Conventional groundwater treatment technology is based on oxidation, and Mn is the last component to be oxidized [5]. The overall potential of Mn^2+^, Fe^2+^ and S^2−^ oxidation by oxygen dissolved in water, calculated from the half-cell oxidation potentials of individual reactions, goes to completion only for sulfides. For both Mn^2+^ and Fe^2+^, the overall potential is negative, and for Mn^2+^, the reaction is quite slow and kinetically unfavorable [6,7]. The subsequent precipitation rate of manganese oxides (MnO_x_) is also slow, particularly when it occurs without a catalyst [8], and the size of the formed MnO_x_ particles depends on the composition of the water [9]. The technological line of groundwater treatment usually consists of an aeration device followed by a filtration section where manganese can be removed in the presence of MnO_x_, especially manganese dioxide (MnO_2_). More importantly, this technology is considered safe for human health and recommended even for mineral water treatment [10]. The MnO_2_ is an element of the filtering bed and plays a crucial role by adsorbing dissolved Mn, which is then oxidized to a solid form [11,12]. Several types of Mn filtering beds are categorized as mineral media with naturally formed in situ MnO_x_ coating, commercial mineral media chemically covered by MnO_x_, and natural manganese ores consisting of MnO_x_ [13]. Mn removal from groundwater is based on heterogeneous catalysis and follows several steps, such as dissolved Mn^2+^ adsorption, and next, its oxidation and precipitation to a solid form of MnO_x_ deposit [12,14,15]. This process is autocatalytic and is present in the Mn^2+^/MnO_2_ systems of Mn removal filter media. However, filtering beds operated for Mn removal from groundwater are characterized by various types of MnO_x_. There is a differentiated degree of crystallinity in the coatings of commercial chemically coated media, and the main identified phases were pyrolusite, ramsdellite, romanechite and hollandite/cryptomelane [16]. Commercial manganese ores for water treatment consist mainly of pyrolusite, ramsdellite and nsutite [17]. A poorly crystalline birnessite is a polymorph often identified in naturally coated filter media [17,18,19]. Each of the oxides is involved in some way in the removal of Mn from the water, although it is not clear how structural differences play a role in this process.

The MnO_x_ solid phase contains a family of MnO_2_ polymorphs that differ in their internal structure (tunnel, layer or spinel) and specific crystallographic phases due to MnO_6_ octahedrons arrangement, as well as in the developing of surface morphological forms and pore dimensions [20]. The different valence state of manganese in MnO_2_ [21]_,_ as well as the ordering of the material structure (amorphous and crystalline) [22], can significantly affect their catalytic activity in chemical and electrochemical reactions [23] and consecutively influence the possibilities of their application for the degradation of organic and inorganic water pollutants [24]. Layered and tunnel MnO_2_ polymorphs are very effective cation exchangers [20,25]. This feature can be used not only in chemical engineering practice but also in environmental engineering areas, such as removing heavy metals and radionuclides from water [26,27] or removing the compounds typical for groundwater, such as Mn and ammonia ions [28,29].

The various properties of MnO_2_ polymorphs (crystallinity and the specific surface area) result in their different adsorption capacities, which were investigated for Pb, Cu and U cationic species [30,31,32], although this trend may also depend on the type of adsorption system, and for Mn, it has not been investigated. In this study, we compared four types of MnO_2_ polymorphs in terms of Mn removal from water. Three tunnel structures of akhtenskite (AKH), cryptomelane (CRY) and pyrolusite (PYR) and one layered birnessite (BIR) were investigated. As mentioned above, pyrolusite and birnessite were chosen because they are the commonly occurring phases in MnO_x_ filter media. Cryptomelane was tested since it can also be present in filter media and is the tunnel analog obtained from layered birnessite [33]. A nano-oxide akhtenskite characterized by high adsorption capacity and obtained in a simple redox reaction [28] was chosen for comparison, especially since this polymorph was studied rarely in the context of water treatment [34]. The scope of the presented research included adsorption tests of the influence of material and solution parameters on the efficiency of manganese removal. The MnO_2_ polymorphs before and after the manganese adsorption were characterized.

## 2. Results and Discussion

### 2.1. Effect of Material and Solution Parameters on Mn Adsorption

The results of Mn adsorption capacity in different pH and ionic strength (I) conditions are presented for AKH and BIR in Figure 1a and for CRY and PYR in Figure 1b,c, respectively. An important observation is a considerable difference between the adsorption capacities of MnO_2_ polymorphs. While AKH and BIR are characterized by comparable and the highest adsorption capacity, the capacity of CRY is from twenty to thirty times lower, and PYR is even about a thousand less (Figure 1a–d). The average values of Mn adsorption capacities at pH = 7 and I = 1 mmol/L were 124 mg/g AKH, 113 mg/g BIR, 4.33 mg/g CRY and 0.069 mg/g PYR, and based on this, the following order can be specified: AKH (ε-MnO_2_) ≈ BIR (δ-MnO_2_) >> CRY (α-MnO_2_) >> PYR (β-MnO_2_). In other research on the role of MnO_2_ polymorphs in adsorption systems, the differences in adsorption capacities were not so substantial. The adsorption of Pb cationic species differed 4–5 times between polymorphs’ types, and the adsorption capacity order was as follows: δ-MnO_2_ > α-MnO_2_ > λ-MnO_2_ > γ-MnO_2_ > β-MnO_2_ [31]. The adsorption of U cationic species varied only marginally (5–15%) between tested MnO_2_ varieties, and finally, the following order was listed: δ-MnO_2_ > α-MnO_2_ > γ-MnO_2_ > β-MnO_2_ [30]. The reasons for the different variability may be both the nature of the adsorbent–adsorbate interactions, such as ion exchange and surface complexation, as well as the properties of the material, e.g., the differentiation in the porous structure of the tested materials, was twenty-fold [31] and six-fold [30]. However, the set of tested polymorphs was not the same in the three analyzed orders, and a coherent part can be distinguished, δ-MnO_2_ > α-MnO_2_ > β-MnO_2_, which confirms the special adsorption features of birnessite compared to cryptomelane and pyrolusite.

All series presented in Figure 1a–c show the differentiation depending on both factors of the water solution. Generally, the lowerpH, the higher solution’s ionic strength and the lower adsorption of Mn. It can be due to the competition of the adsorbate cation with other cations, including hydronium ions, and the electrostatic repulsion between the protonated surface of the adsorbent and the adsorbate cation [28,30,35]. The special feature of CRY and PYR polymorphs is the lack of stability in an acidic environment (Figure 1b,c). In both cases, a pH less than six caused the release of Mn from the oxide and an increase in its concentration in the solution. It was especially intensive for CRY. However, this result has great importance for the exploitation of manganese ores in Mn removal filters because these materials often contain CRY and PYR polymorphs [17,36].

The variance tests were analyzed to confirm the relationships described above for the data population. Because adsorption did not occur in the case of Mn release, the negative q values were replaced by values close to zero for analysis purposes. Analysis with the Kruskal–Wallis H test showed a significant difference in adsorption capacity level between the MnO_2_ type, H = 294.32; *p* = 0.0001 (Appendix A). The post hoc test indicated that significant differences (*p*-value < 0.05) exist between BIR, AKH and CYR, PYR polymorphs (Figure 1d, Appendix A). In contrast, there are no statistically significant differences in adsorption capacity between BIR and AKH or CYR and PYR. The Kruskal–Wallis H test showed a significant difference in adsorption level between the separate groups according to initial pH, H = 78.6; *p* = 0.00001 (Appendix A). The post hoc test indicated in most cases that significant differences (*p*-value < 0.05) exist between groups of pH 2–5 and groups of pH 6–9 (Figure 1e, Appendix A). These results are confirmed in operational practice at water treatment plants, where the recommended pH for manganese ore filter media is higher than 6.5 or 7.0 [13]. In groups with an initial level of pH 2–5, there are no significant statistical differences in adsorption capacity (*p*-value > 0.05), and similarly in groups with an initial level of pH 6–9 (*p*-value > 0.05). In the case of ionic strength, the Kruskal–Wallis H test showed no statistical significance in adsorption capacity level between the separate groups, H = 1.55; *p* = 0.67 (Figure 1f, Appendix A). The ionic strength of the solutions used in the experiment corresponds to a low to medium salinity of the water [37], which means that the variability of this parameter has no considerable impact on the technological effects of Mn removal on filter media containing MnO_2_.

The effect size of the relationship between Mn adsorption capacity and the factors groups was determined based on the etha-squared estimate (η^2^) as well as the epsilon-squared coefficient (ε^2^). The η^2^ indicates the percentage of the variance of the dependent variable explained by the independent variable, and ε^2^ assumes the value from 0 (indicating no relationship) to 1 (indicating a perfect relationship between the dependent and independent variable) [38]. The calculated values of η^2^ and ε^2^ for the relationship between Mn adsorption capacity and MnO_2_ polymorph type were 76.7% and 0.768, respectively. At the same time, the estimates for the relation between Mn adsorption capacity and pH of the solution were 19.0% and 0.205. The results directly indicate a nearly four times stronger relationship between adsorption level and MnO_2_ type than adsorption level and initial pH. Statistical analysis proves that the type of MnO_2_ has the strongest influence on manganese adsorption from water solution, with amorphous BIR and AKH having the highest performance in this regard. Many authors indicate a crucial role of birnessite in adsorption systems due to its high adsorption and redox reactivity and significant surface area with the presence of hydroxyl and other oxygen-containing groups [30,31,39]. At present, a combination of Fe and Mn removal in flow reactors with membrane filtration is an integrated technology with great flexibility and a future-oriented direction for a decentralized water supply [40,41]. It seems that using a selected variety of powdered MnO_2_ in flow reactors with membrane could be important for increasing the efficiency of the systems for Mn removal from groundwater. In this sense, the research on the factors that influence the Mn removal process and the MnO_2_ polymorphs’ role has not only a cognitive sense but also a practical dimension.

### 2.2. Influence of Mn Adsorption on MnO_2_ Polymorphs’ Properties

The characteristics of tested materials are presented in Table 1 and Figure 2 and Figure 3 as well as in Appendix A. Based on XRD results, it is clear that prepared materials are single-phase without crystalline amendments. The tunnel polymorph AKH is poorly crystalline and contains a high proportion of the amorphous phase, confirmed by a distinctive spectrum (Figure 2a). The layered polymorph BIR is of the same amorphous state (Figure 2b), and both contain small-size crystallites. The two tunnel MnO_2_ polymorphs, CRY and PYR, are strongly crystalline with well-observed reflexes on the XRD diffractogram. The highly developed crystallites with large dimensions are present, especially in the PYR sample. The morphology of the AKH polymorph is characterized by spherical particles forming larger spherical clusters (Figure 3a). This microphotography shows the highly developed AKH surface structures correlate with the microporosity of this material, confirmed by ASAP analysis (Table 1). The MnO_2_ nano-polymorphs are characterized by a larger specific surface area (SSA), lower density and increased reactivity [42], and only AKH from the tested materials can be described as a nanomaterial. The obtained results are consistent with others, where depending on the structural cation or preparation time, the AKH characteristics were as follows SSA 239–416 m^2^/g, pore volume 0.35–0.57 cm^3^/g, and the pore diameter 4.0–6.2 nm [28,43]. The surface morphology of BIR (Figure 3c) shows the typical ball-flower structures also observed by others for this type of MnO_2_ polymorph [31,44]. The porous structure of the obtained BIR sample is average compared to the reported SSA in the range of 4.56–270.8 m^2^/g [31,45]. The thermal treatment converted the BIR sample into the CRY with a one-third loss of the SSA and one-fourth of the pore volume. However, BIR and CRY samples are similar in surface development compared to the other polymorphs studied. The tiny rods of crystals typical for cryptomelane [46] are documented in the CRY sample (Figure 3e). The XRD diffractogram and the SEM image of the PYR sample indicate its strong crystallization and very low surface morphological differentiation (Figure 2d and Figure 3g). This observation is in agreement with the results of porosimetry analysis, confirming the marginal values of SSA and pores volume.

When analyzing the results of XRD, the adsorption of Mn does not create any additional crystalline or poorly crystalline phases in the tested polymorphs (Figure 2). The Mn deposition substantially affects the characteristics of AKH and BIR polymorphs due to the high and similar adsorbate loading. However, the changes are not of the same nature. The total pore volume and the SSA of the AKH sample decreased by 25% and 28%, respectively, but the most significant changes occurred in the micropores, as their volume and SSA decreased by 76% and 77%, respectively. The Mn adsorbate is well visible in the SEM micrograph (Figure 3b), and the spherical particles of the original material appear to be immersed in it. By using the EDS analysis (Appendix A), it can be noted that the surface of the raw AKH is dominated by O and Mn with a small admixture of Ca, while the formed adsorbate layer contains only O and Mn, and the signal from Ca is no longer intense. The presence of the adsorbate does not affect the proportion between the amorphous and crystalline phases of the AKH sample. Generally, the intensity of reflexes in the samples after Mn adsorption decreases except for the BIR. Additionally, in the diffractogram of BIR after Mn adsorption (Figure 2b), a hump also originating from the birnessite phase is visible near 55° 2θ. This indicates that the structure of birnessite is extended after the adsorption process. It is in agreement with porosimetry analysis results, showing an intensive increase in the SSA and the pore volume of BIR with Mn adsorbate and simultaneously a decrease in the average diameter of pores. The surface of BIR before and after Mn adsorption (Figure 3c,d) consists of similar plate-like forms, but the material with adsorbate is significantly more amorphous. The elements present on the surface of the BIR sample before and after adsorption are O, Mn and K (Appendix A), characteristic of the birnessite phase. It can indicate the development of BIR structure and the partially catalytic nature of Mn removal on this polymorph. According to other studies, this polymorph, after the adsorption of divalent Mn, can be transformed into trivalent Mn-rich BIR, triclinic BIR or tunnel structure according to the presence of different countercations [47], but the research was performed in an anoxic environment so the analogy cannot be direct. Indirect confirmation may also be that the AKH and BIR are characterized by comparable adsorption capacity, although AKH has a ten times higher SSA than BIR. For this reason, the SSA seems to be not always a useful parameter for Mn^2+^/MnO_2_ system. Other researchers consider the SSA as a significant predictor of the adsorption capacity of MnO_2_ with different crystallographic phases, although this applies to Pb^2+^/MnO_2_ and U^6+^/MnO_2_ adsorption systems [30,31]. The results of porosimetry, XRD and SEM analyses indicate well that the low loading of Mn adsorbate does not change the characteristics of highly crystalline CRY and PYR polymorphs. Similarly, the type of elements on the CRY and PYR surfaces does not change as a result of Mn adsorption, and they are O, Mn and K and O and Mn, respectively (Appendix A). The statistical analysis provides proof that the polymorphs’ type is the most important factor for efficient Mn removal from water solution. The results shown in Section 2.1 (Figure 1d) indicate the strongly highest adsorption capacities of amorphous MnO_2_ polymorphs (AKH and BIR) compared to the crystalline (CRY and PYR). This conclusion is consistent with the results of others, proving that the amorphous nature of MnO_2_ relates to more oxygen vacancy defects on its surface and higher catalytic activity [30,48].

## 3. Materials and Methods

### 3.1. Chemicals

The following chemicals of analytical grade purchased from Chempur (Piekary Śląskie, Poland) were used in the experiments: MnSO_4_·H_2_O, MnCl_2_·4H_2_O, NaHCO_3_, Mn(NO_3_)_2_·4H_2_O, KMnO_4_, NaCl, HCl of 35–38%, H_2_SO_4_ of 98%, HNO_3_ of 65% and NaOH. The TraceCERT^®^ manganese standard was used for analysis with a flame atomic absorption spectrometer. Double-distilled water was used throughout.

### 3.2. Preparation of MnO_2_

Four types of MnO_2_, akhtenskite (AKH), birnessite (BIR), cryptomelane (CRY) and pyrolusite (PYR), were prepared. AKH was synthesized by simple co-precipitation of MnCl_2_ and KMnO_4_ [28]. According to the methodologies described by [33], BIR was prepared in a hydrothermal method based on HCl and KMnO_4_ reaction, and the part of the product was next ignited at 400 °C to obtain CRY. PYR was synthesized by the thermal decomposition of Mn(NO_3_)_2_ at 180 °C. All oxides were washed until the conductivity of double-distilled water stopped changing. In the final step, the samples were dried at 100 °C and next ground in the agate mortar.

### 3.3. Batch Experiment of Mn Removal

The test solution of Mn was prepared according to [49] using MnSO_4_·H_2_O and NaHCO_3_ salts. The effect of solution properties and MnO_2_ type on the adsorption of divalent Mn was tested in the range of pH 2–9, ionic strength (I) of 1.0–50 mmol/L, constant initial concentration (C_i_) 10 mg Mn/L and contact time 24 h. The Mn adsorption capacity (q) was calculated as q = (C_i − f_)·V/m, where C_f_ is Mn concentration in the solution after adsorption, V is the solution volume and m is the mass of MnO_2_. The ionic strength of the test solution was increased by adding NaCl, and pH was adjusted by NaOH and HCl. The doses of MnO_2_ were 0.05 g/L (AKH, BIR), 0.2 g/L (CRY) and 1.0 g/L (PYR). The analytical balance Kern ABT 220-4M (Kern&Sohn GmbH, Balingen, Germany) was used for weighting. All tests were performed in glassware by shaking and incubating at 10 °C, a typical temperature for groundwaters. The experiment was performed in triplicate. The test solution (after adsorption) was separated from the MnO_2_ mixture using MCE 0.22 μm membrane syringe filters (Alchem, Toruń, Poland). The Mn concentration in the solution was analyzed using a flame atomic absorption spectrometer iCE 3000 series (ThermoScientific, Waltham, MA, USA) in an acetylene–air mixture at wavelength 279.5 nm and bandwidth 0.2 nm, achieving a sensitivity of 0.02 mg/L and detection limit 0.0016 mg/L. The pH was analyzed using gel probes IntelliCAL working with HQ40d meter (Hach, Loveland, CO, USA).

### 3.4. Data Analysis

A statistically significant difference between analyzed factors groups (polymorph type, pH and ionic strength of solution) in the Mn adsorption capacity (q) was determined using an analysis of variance (ANOVA, Cambridge, UK) test in STATISTICA 13.3 software (TIBCO Software Inc., Palo Alto, CA, USA). The non-parametric Kruskal–Wallis H test was applied because a homogeneity of variance and normal distribution of a dependent variable (q) in all separated groups of analyzed factors (in most cases, the Shapiro–Wilk test *p*-value was less than the determined significance level α = 0.05) were not fulfilled [50]. A post hoc test for non-parametric data was used to provide insight into exactly which groups differed from each other. The effect size of the relationship between the analyzed dependent and independent variables (Mn adsorption and the factors groups, respectively) was estimated using the Kruskal–Wallis H test based on two measures (η^2^) and (ε^2^) [38].

### 3.5. Characterisation of MnO_2_

The synthesized MnO_2_, as well as the samples after adsorption (pH = 7.0, I = 1.0 mmol/L), were characterized as follows. The specific surface area (SSA), volume and average diameter of pores were measured by low-temperature nitrogen adsorption using analyzers ASAP 2020C and ASAP 2420M (Micromeritics, Norcross, GA, USA), and before analysis, the samples were degassed at 100 °C. The SSA was calculated using the Brunauer–Emmett–Teller method. The morphology of the oxides was analyzed with a field emission scanning electron microscope (FE-SEM) Quattro S (Thermo Fisher Scientific, Waltham, MA, USA). The micrographs were taken using an Everhart–Thornley detector with a voltage of incident beam of 20 kV. Samples were coated with a thin layer of carbon <10 nm, followed by Pt-plasma sputtering up to a dozen nm because of the tendency to charge. The elemental composition of polymorphs’ surfaces coated with carbon was analyzed using energy dispersive X-ray spectroscopy (EDS) detector Octane Elect Plus (EDAX Inc., Pleasanton, CA, USA). The crystal structure of materials was analyzed using an X-ray diffractometer (XRD) D8 Advance (Bruker, Billerica, MA, USA) with Cu Kα (λ = 0.154056 nm) radiation, with scan step 0.015° 2θ, scan rate 2 s/step, and scan range from 15 to 95° 2θ, operated at 40 kV and 40 mA. The powder was placed on Si low background sample holder at 15 rpm. The average crystallite sizes (D) samples were calculated from the XRD line widths by applying the well-known Debye–Scherrer Equation D = 0.89*D* = 0.89λ/Bcosθ, where λ is the wavelength of the X-ray in nanometres, B is the peak width at half-height, and θ is the angle between the incident and diffracted beams in angular degrees.

## 4. Conclusions

Statistical analysis showed that the type of MnO_2_ polymorph is the most significant factor in removing Mn from an aqueous solution. The relationship between the adsorption level and pH of the solution is four times weaker in the range of 2–9, and there is no significant difference in adsorption capacity in various ionic strength of the solution in the range of 1–50 mmol/L. The poorly crystalline polymorphs, AKH and BIR, are stable in the aqueous environment in the range of 2–9 pH and ionic strength of 1–50 mmol/L, whereas the highly crystalline CRY and PYR show instability in acidic conditions at pH below 6, which causes in manganese leaching from the oxides. The tunnel AKH and the layered BIR are characterized by comparable and the greatest adsorption capacity, twenty to thirty times higher than the capacity of the tunnel CRY and a thousand higher than the capacity of the tunnel PYR. The order of adsorption capacity is determined as follows: AKH (ε-MnO_2_) ≈ BIR (δ-MnO_2_) >> CRY (α-MnO_2_) >> PYR (β-MnO_2_); because of the average Mn adsorption capacities: 124 mg/g AKH, 113 mg/g BIR, 4.33 mg/g CRY and 0.069 mg/g PYR. The high Mn adsorbate loading differently affects the AKH and BIR surfaces. The micropores of AKH are filled by the adsorbate, which results in a decrease in their volume and the size of specific surface area. Conversely, the adsorption of divalent Mn leads to the development of the specific surface area and porosity of the BIR, and the formed adsorbate follows the origin BIR structure but with greater amorphity. The low adsorbate loading does not change the highly crystalline CRY and PYR.

## Figures and Tables

**Figure 1 ijms-24-04448-f001:**
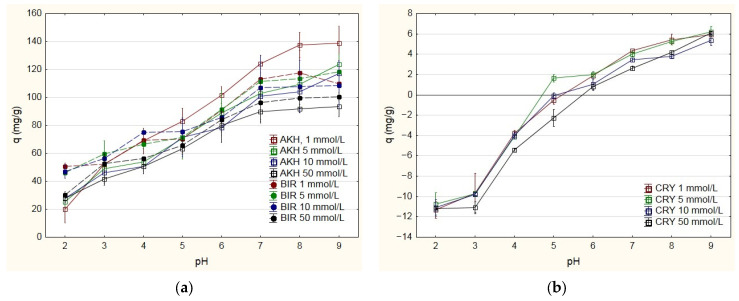
Mn adsorption capacity in different conditions of pH and ionic strength for the following: (**a**) akhtenskite (AKH) and birnessite (BIR); (**b**) cryptomelane (CRY); (**c**) pyrolusite (PYR). The ANOVA results for: (**d**) manganese dioxide (MnO_2_) polymorph type; (**e**) pH of water solution; (**f**) ionic strength of water solution.

**Figure 2 ijms-24-04448-f002:**
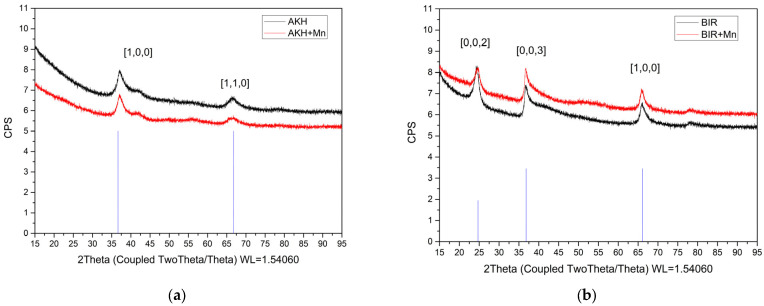
The diffractograms of MnO_2_ polymorphs before and after Mn adsorption: (**a**) AKH and AKH + Mn; (**b**) BIR and BIR + Mn; (**c**) CRY and CRY + Mn; (**d**) PYR and PYR + Mn. The XRD JCPDS codes are as follows: (**a**) 00-030-0820, (**b**) 00-013-0105, (**c**) 00-004-0603 and (**d**) 00-024-0735.

**Figure 3 ijms-24-04448-f003:**
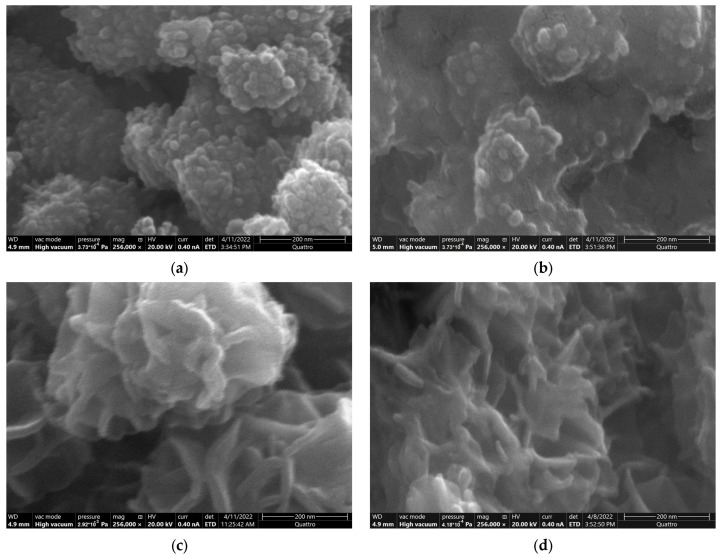
SEM microphotographs of MnO_2_ polymorphs’ surfaces before and after Mn adsorption: (**a**) AKH; (**b**) AKH + Mn; (**c**) BIR; (**d**) BIR + Mn; (**e**) CRY; (**f**) CRY + Mn; (**g**) PYR; (**h**) PYR + Mn.

**Table 1 ijms-24-04448-t001:** Characteristics of manganese dioxide (MnO_2_) polymorphs before and after manganese (Mn) adsorption.

Parameters	AKH	AKH + Mn	BIR	BIR + Mn	CRY	CRY + Mn	PYR	PYR + Mn
XRD results								
identified phases	akhtenskite	akhtenskite	birnessite	birnessite	cryptomelane	cryptomelane	pyrolusite	pyrolusite
percentage of crystalline phase (%)	65.1	65.5	60.4	64.6	88.6	87.6	85.7	85.3
percentage of amorphous phase (%)	34.9	34.5	39.6	35.5	11.4	12.4	14.3	14.7
average crystallite size (nm)	5.7	7.3	8.1	7.2	31	35	72	68
Porosimetry results								
specific surface area (m^2^/g)	334.9	240.6	34.12	79.99	23.40	22.48	0.1726	0.0976
specific surface area of micropores (m^2^/g)	117.3	27.98	-	-	-	-	-	-
total volume of pores (cm^3^/g)	0.3555	0.2649	0.1463	0.1962	0.1117	0.1123	0.0008	0.0005
volume of micropores (cm^3^/g)	0.0522	0.0122	-	-	-	-	-	-
average diameter of pores (nm)	3.8	5.5	17	8.6	19	20	20	25

## Data Availability

The data presented in this study are openly available in RepOD at https://doi.org/10.18150/96JKV0 (accessed on 19 February 2023).

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
