# Peer review of "Significance of MnO_2_ Type and Solution Parameters in Manganese Removal from Water Solution"

_ijms, 2023, doi:10.3390/ijms24054448_

Round 1

Reviewer 1 Report

Background Remarks:

The experimental setup is hardly described before the results are stated. When reading the results, it is impossible to understand the results without knowing how the experiments were conducted: (temperature conditions, solid/liquid ratio, duration of the reactions,...).
The materials and methods should be placed before the results.
This point must imperatively be corrected before a new submission.
For the remaining part, the paper brings new scientific and technical knowledge that advances the knowledge.
I recommend a new submission of the paper after reorganization of the structure.

Detailed remarks:
Lines 38-39: the list of problems related to the presence of manganese is long, you can add that the presence of manganese stains the laundry!
Line 45: give more bibliographic references on the slow kinetic character of manganese oxidation. The difficulty of demanganification is more related to the slow kinetics of precipitation than to the kinetics of oxidation.
Lines 51 -52: the catalytic role of the beds (especially the role of manganese minerals) on the kinetics of manganese precipitation should be developed.
Lines 70-72: these lines appear to be irrelevant.

Author Response

Dear reviewer,

Thank you for giving us the opportunity to submit a revised draft of our manuscript. We appreciate the time and effort that you have dedicated to providing your valuable feedback on our manuscript. We are grateful to you for your insightful comments on our paper. We have been able to incorporate changes to reflect most of the suggestions provided by you. We have highlighted the changes within the manuscript. Here is a point-by-point response to your comments and concerns.

Question 1: The experimental setup is hardly described before the results are stated. When reading the results, it is impossible to understand the results without knowing how the experiments were conducted: (temperature conditions, solid/liquid ratio, duration of the reactions, ...). The materials and methods should be placed before the results. This point must imperatively be corrected before a new submission.

For the remaining part, the paper brings new scientific and technical knowledge that advances the knowledge.
I recommend a new submission of the paper after reorganization of the structure.

Answer: Dear reviewer, it may be better to write the experimental conditions before the results as you mentioned, but we have to comply with the journal rules. According to the journal template, we had to write results and discussion section before the material and methods (here you can find journal template and here about journal structure). We asked the editor of the journal about the possibility of changing the manuscript’s order and the editor emphasized avoiding changing the order. Therefore, we cannot change the structure of our manuscript.

Detailed remarks:

Question 2: Lines 38-39: the list of problems related to the presence of manganese is long, you can add that the presence of manganese stains the laundry!

Answer: We added it into the text (line 40).

Question 3: Line 45: give more bibliographic references on the slow kinetic character of manganese oxidation. The difficulty of demanganification is more related to the slow kinetics of precipitation than to the kinetics of oxidation.

Answer: This sentence contained short explanation and we extended our thought by adding more information about the reasons of why Mn oxidation is difficult (lines 46-53).

Question 4: Lines 51 -52: the catalytic role of the beds (especially the role of manganese minerals) on the kinetics of manganese precipitation should be developed.

Answer: we developed the role of Mn minerals on the kinetics of Mn precipitation (lines 61-64)

Question 5: Lines 70-72: these lines appear to be irrelevant.

Answer: Thank you for finding this subtle gap. We agree there was no logical sequence in this paragraph. We reedited this part (lines 78-87).

Reviewer 2 Report

The manuscript titled ‘’Significance of MnO2 type and solution parameters in manganese removal from water solution’’ by Michel et al. describes elaborate studies of Mn adsorption properties of the different polymorphs of MnO2 in water. Quantitative studies of the impact of different polymorph types (akhtenskite ε-MnO2, birnessite δ-MnO2, cryptomelane α-MnO2 and pyrolusite β-MnO2), pH (2-9) and ionic strength (1-50 mmol/L) of solution on the adsorption of Mn was investigated here. Finally, the authors try to explain the reason behind the different adsorption by relating to the morphology and crystallinity of different polymorphs. The results are within the scope of the study considering the application perspective. However, some of the comments need to be addressed.

  1. Schematic structure of crystalline phases of different polymorphs can be given by Single crystal analysis if possible, in the main paper so that it may be clear to the authors.
  2. A comparitive discussion relating the Mn adsorption results with crystallinity and amorphous nature of different polymorphs may be well and good to incorporate in section 2.2
  3. Some of the abbreviations eg. SSA on page 5 line 190 not elaborated.
  4. The statement on page 5 line 190 ‘’The thermal treatment of the BIR sample became it into the CRY with a loss of one-third of the SSA and one-fourth of the pore volume’’ is not clear. Is it like BIR gets converted to CRY?
  5. The SEM images in Fig 3 look so insignificant without any much difference before and after Mn Adsorption. An EDAX study is suggested showing a quantitative analysis of Free Mn adsorbed in different polymorphs. 

Author Response

Dear reviewer,

Thank you for giving us the opportunity to submit a revised draft of our manuscript. We appreciate the time and effort that you have dedicated to providing your valuable feedback on our manuscript. We are grateful to you for your insightful comments on our paper. We have been able to incorporate changes to reflect most of the suggestions provided by you. We have highlighted the changes within the manuscript. Here is a point-by-point response to your comments and concerns.

Reviewer comments:

The manuscript titled ‘’Significance of MnO2 type and solution parameters in manganese removal from water solution’’ by Michel et al. describes elaborate studies of Mn adsorption properties of the different polymorphs of MnO2 in water. Quantitative studies of the impact of different polymorph types (akhtenskite ε-MnO2, birnessite δ-MnO2, cryptomelane α-MnO2 and pyrolusite β-MnO2), pH (2-9) and ionic strength (1-50 mmol/L) of solution on the adsorption of Mn was investigated here. Finally, the authors try to explain the reason behind the different adsorption by relating to the morphology and crystallinity of different polymorphs. The results are within the scope of the study considering the application perspective. However, some of the comments need to be addressed.

Question 1: Schematic structure of crystalline phases of different polymorphs can be given by Single crystal analysis if possible, in the main paper so that it may be clear to the authors.

Answer: All samples examined are powder samples. The powder was placed on Si low background sample holder and then measured with the following parameters: 15-95 2theta, sample rotation (including holder) 15 RPM, increment 0.015 2theta, step time 10s. XRD diffractometer is a Bruker D8 Advance with CuKα (λ = 0.154056 nm) radiation, operated at 40 kV and 40 mA. This diffractometer is dedicated to powder measurements and it is not possible to perform standard Single Crystal measurements on it. In powder diffraction, all the symmetry-equivalent reflections have the same d spacing with the result that individual intensities cannot be measured. That is why we focused in the manuscript (XRD part) on phase analysis. We tried to clear more about our procedure in the section 3.5 (lines 342-344).

Question 2: A comparative discussion relating the Mn adsorption results with crystallinity and amorphous nature of different polymorphs may be well and good to incorporate in section 2.2

Answer: The discussion about Mn adsorption results of different types of polymorphs were added in the section 2.2 (lines 266-272).

Question 3: Some of the abbreviations eg. SSA on page 5 line 190 not elaborated.

Answer: We added the full name (line 204).

Question 4: The statement on page 5 line 190 ‘’the thermal treatment of the BIR sample became it into the CRY with a loss of one-third of the SSA and one-fourth of the pore volume’’ is not clear. Is it like BIR gets converted to CRY?

Answer: Thank you for the advice. We changed the sentence (line 212).

Question 5: The SEM images in Fig 3 look so insignificant without any much difference before and after Mn Adsorption. An EDAX study is suggested showing a quantitative analysis of free Mn adsorbed in different polymorphs.

Answer: Yes, we agree with the Reviewer. The SEM images are quite the same except for akhtenskite. We underlined it in the manuscript. We added the results of SEM-EDS as a supplementary file. Also the comments have been inserted to paragraph 2.2. We could deduct, that no additional elements have been deposited on the polymorphs’ surfaces. In our opinion, the results of the quantitative analysis are not directly useful because of the analysis uncertainty on the level of 10% (lines 191-192, 237-240, 250-251, 264-266, 338-341).

Moderate English changes were applied for the manuscript.

Reviewer 3 Report

This paper study significance of manganese oxides type and solution parameters for removing manganese from water solution. The impact of manganese oxides polymorph type, solution pH and ionic strength impact on Mn adsorption level was investigated. The studies were quite systematic and the resulted were well organized by the authors. I’d like to recommend the publication of this paper in molecules after revision.

1. Authors should show EDX results to support manganese oxides exist

2. Authors should show FTIR results to explain whether manganese oxides surface structure had a influence Mn adsorption.

3. Authors should show XRD JCPD code on XRD results.

Author Response

Dear reviewer,

Thank you for giving us the opportunity to submit a revised draft of our manuscript. We appreciate the time and effort that you have dedicated to providing your valuable feedback on our manuscript. We are grateful to you for your insightful comments on our paper. We have been able to incorporate changes to reflect most of the suggestions provided by you. We have highlighted the changes within the manuscript. Here is a point-by-point response to your comments and concerns.

Reviewer comments:

This paper study significance of manganese oxides type and solution parameters for removing manganese from water solution. The impact of manganese oxides polymorph type, solution pH and ionic strength impact on Mn adsorption level was investigated. The studies were quite systematic and the resulted were well organized by the authors. I’d like to recommend the publication of this paper in molecules after revision.

Question 1: Authors should show EDX results to support manganese oxides exist.

Answer: We added the results of SEM-EDS as a supplementary file. In addition, the comments have been inserted to the paragraph 2.2 (lines 191-192, 237-240, 250-251, 264-266, 338-341). We could deduct, that no additional elements have been deposited on the polymorphs’ surfaces. Of course, it is very difficult to distinguish the original Mn and O elements and the deposited Mn and O in the Mn/MnO2 system. The isotopic labelling is needed for obtaining the answer, but we could not do this type of research.

Question 2: Authors should show FTIR results to explain whether manganese oxides surface structure had an influence Mn adsorption.

Answer: We agree with the Reviewer that FTIR results are really helpful in the material analysis. Unfortunately, we could not do this measurement for our samples. We hope the Reviewer can understand our explanation.

Question 3: Authors should show XRD JCPD code on XRD results.

Answer: We added information about JPDS codes (Fig. 2). As well as that, we added more information about measurement parameters in Section 3.5 (lines 342-344)
